# Does the Type of Smartphone Usage Behavior Influence Problematic Smartphone Use and the Related Stress Perception?

**DOI:** 10.3390/bs12040099

**Published:** 2022-04-09

**Authors:** Lea-Christin Wickord, Claudia Michaela Quaiser-Pohl

**Affiliations:** Institute of Psychology, University Koblenz-Landau, 56070 Koblenz, Germany; quaiser@uni-koblenz.de

**Keywords:** problematic smartphone use, perceived stress, types of smartphone use

## Abstract

Problematic smartphone use (PSU) is defined as the inability to control the time spent on smartphones, which has long-term negative impacts on daily life. The use-and-gratifications approach is applied to smartphones and describes the extent to which users devote themselves to smartphones to obtain gratifications. These gratifications can be represented in the types of use (process, social, and habitual). This study examines the associations between PSU and the different types of use and their effects on perceived stress and self-perceived PSU. *N* = 108 subjects participated (65 women, 41 men, 2 diverse, mean age = 31.8; range 17–70). They completed the Mobile Phone Problem Use Scale (MPPUS-19), Perceived Stress Scale (PSS-10), a questionnaire on types of use, and a self-created scale for self-perceived PSU. Multiple linear regressions and correlations were calculated and show a relationship between PSU and perceived stress. All three types of use were shown to be predictors of PSU. For stress perception, only process use is a predictor. Both PSU and stress perception are predictors of the self-perceived PSU. Both stress and PSU interact with each other, and the different types of use determine how stressful smartphone use is perceived to be.

## 1. Introduction

In recent years, the smartphone has become a constant companion for many people and has thus taken on a central role in everyday life [1]. According to calculations by the agency Newzoo, there were approximately 3 billion smartphone users worldwide in 2018 [2]. The special feature of the smartphone lies on the one hand in the combination of mobility and versatility of applications and uses, and on the other hand in the fact that the smartphone combines several devices in one [3,4,5]. Thus, users can both access news and information from the Internet and communicate with their social contacts all at once [4,6]. At the same time, technology is continuously evolving. Due to its handy size, the smartphone can always be carried close to the body and is quickly at hand, for example, to answer text messages, search the Internet for specific information, or play a game [7,8,9,10,11]. Due to the increasing integration of the smartphone into more and more areas of life, the duration of use is also increasing [4,12,13] and the smartphone is becoming an essential part of everyday life [14], but what are the motives for this permanent use of the smartphone?

### 1.1. Theories on the Motives for Smartphone Use

#### 1.1.1. Uses-and-Gratifications Approach

The uses-and-gratifications approach is a model of media use research and examines or describes the extent to which recipients devote themselves to certain media offerings to derive certain benefits from the respective media. The approach focuses on gratifications (satisfaction of needs) that result from the use of the chosen offers and can thus be added as motivation-theoretical aspects to represent media use and media effects [15]. Meanwhile, users are considered active, goal-oriented subjects who have individual needs and expectations for specific media offerings [16]. The following needs or motives exist for media use: 1. the need for information (orientation, seeking advice, learning), 2. the need for entertainment (escapism, relaxation, sexual stimulation), 3. the need for personal identity (search for models of behavior, reinforcement of personal values), and 4. the need for integration and social interaction (substitute for sociability, role model, conversation) [17].

#### 1.1.2. Compensatory Internet Use Theory (CIUT)

Kardefelt-Winther’s [18] compensatory internet use theory (CIUT) can be seen as a further development of the uses-and-gratifications approach, which can also be applied to smartphone use [19]. It seeks to understand negative life events and stressors that drive users to use technology excessively. For example, the motivation may be to mitigate the negative emotions associated with the stressors. By illuminating the relationship between mental health and problematic smartphone use, conclusions can be drawn about the extent to which online activities can act as compensators for psychosocial problems even if it involves negative consequences. The core of the theory is that the problem is the individual’s reaction to negative situations, the resolution or termination of which is enabled by the use of technology. For example, if a person lacks social contact, he feels the need for social interaction. This is provided to him by certain smartphone applications (e.g., social media) and can have both positive and negative effects: Positive, because the need for social contact has been gratified, and negative, because digital exchanges cannot adequately replace real ones and, depending on various factors (such as the presence of certain psychosocial problems), the person runs the risk of becoming dependent on the Internet or smartphone use for his or her need to satisfy social stimulation, which in turn can lead to excessive use and possibly problematic usage behavior [18]. The compensatory internet use theory states that problematic use can arise as a consequence of negative feelings or problems. Additionally, in a further step, there is more talk of reciprocal interaction, in which the smartphone is used to escape or alleviate negative feelings. However, excessive use then leads to increased negative feelings and stress as a result [20]

Apart from the motives and gratifications, however, increasing usage has negative consequences: Research findings in recent years show that device use may be associated with a variety of psychological and physiological problems caused by problematic smartphone use, which leads to more stress [12,20].

### 1.2. Problematic Smartphone Use

In the course of this, the construct of problematic smartphone use (PSU) became established. Problematic smartphone use behavior is predominantly conceptualized in scientific discourse as behavioral addiction (substance-independent dependence) and is distinguished from non-problematic use behavior by symptoms such as withdrawal symptoms (when the device is unavailable), tolerance development (use must be intensified to obtain the same level of gratification), dangerous use (e.g., while driving), and conflicts with the social environment due to the use behavior [19,21]. In addition, technological triggers (such as notification tones), a variety of application options, and the constant tangibility of the device are also facilitating factors [12,19,22,23]. Measurement tools are often modeled on or have evolved from, the concepts of gambling and computer game addiction. The concept of internet addiction is also related to this [24].

Billieux [25] defines problematic smartphone use as “an inability to regulate one’s use of the smartphone, which eventually involves negative consequences in daily life”. As Kuyulu and Beltekin [4] describe: “The use of smartphones both gives pleasure to the person as a result of use and saves them from pressure or anxiety. Such reinforcement makes it easy to be addicted to the smartphone”. Even though the symptoms of problematic smartphone use are similar to the symptoms of other addictions, it is essential to consider problematic smartphone use separately from addictions according to the ICD-10 criteria. The consequences of problematic smartphone use are not comparable to the intensity and limitations associated with other addictions [14]. Long et al. [26] therefore use the term problematic smartphone use instead of an addiction term so that the phenomenon escapes automatic pathologization. This term will continue to be used in this study.

To clarify the intensity and severity of problematic smartphone use, it can be differentiated between excessive-functional media use, which is characterized by users implementing a request in a goal-directed and self-controlled manner and not experiencing serious negative consequences; excessive-dysfunctional media use, which is characterized by a low level of goal-directedness and control and is less effective, but is not experienced as burdensome by users; and dependent media use, in which action control is even lower than in the other forms and the individual perceives the usage behavior or the usage time itself as inappropriately high [27].

There has been a growing body of research on the psychopathological factors associated with problematic smartphone use over the past 10 years [28]. Among the most commonly studied associations are those between depression, anxiety, and problematic smartphone use [19,29,30,31,32]. People with depression use their smartphones to cope with or suppress and avoid their depressive, negative emotions [20]. The resulting excessive smartphone use leads to increased sleep problems and stress [31], which in turn increases depressive symptoms and leads to a vicious cycle [33,34,35]. Other studies on the relationship between psychological factors and smartphone use behavior or the propensity to engage in problematic use behavior found positive associations between problematic smartphone use and technostress in several studies [36,37].

### 1.3. Type of Use as a Factor for Problematic Smartphone Use

Another factor related to problematic smartphone use is the type of use and how the different types of use provide gratification to the individual. A general distinction is made between social, process, and habitual use. Social use includes interaction with one’s social network via social media and instant messaging and satisfying the need for social interaction. Process use, on the other hand, describes content-related media consumption such as reading news websites, streaming videos, or playing in-app games, thus satisfying the expectation to pursue enjoyable activities [22,38,39]. Habitual use can be defined as a habitual behavior as an automatic response to certain stimuli coming from within, e.g., a certain craving or emotion, or from outside, such as by a ringtone or a smartphone screen lighting up [40], leading to the automatism of unlocking the phone to check for new notifications [41]. If this behavior leads to desirable outcomes, it is performed more often and habitual use increases due to gratification [22,42].

Studies have found either greater associations with habitual and process smartphone use [22] or social use [23] and problematic use behaviors. It is unclear to what extent the type of use influences specific types of problematic use. Last but not least, stress and stress perception play a role in the context of problematic smartphone use and can be correlated with the type of use [31].

### 1.4. Stress and Coping Strategies

According to the transactional approach, stress does not result predominantly from the external stimuli acting on a person but must be considered concerning the individual’s perception, appraisal of a situation, and the coping strategies available for its coping [43,44]. Thus, coping strategies can be generally divided into adaptive (or functional) and maladaptive (or dysfunctional) strategies [45]. In the context of the transactional stress model, the various problem-oriented and emotion-oriented coping strategies can now also be rated as adaptive or maladaptive, depending on their success, with problem-oriented strategies often categorized as adaptive (because they are active) and emotion-oriented strategies categorized as maladaptive because they do not solve the problem [46].

Stress perception has been investigated in previous studies mainly concerning Internet addiction [47,48]. Here, significant positive associations were found. In recent years, research on stress perception has been extended to include the factor of smartphone use or problematic smartphone use. The review study by Elhai et al. [19] confirms significant positive associations between higher stress perception and problematic smartphone use in five of the total six studies. On the one hand, a longitudinal study by Thomée et al. [31] was able to show that individuals categorized as heavy users of computers, social media, and smartphones had higher rates of stress, depression, and sleep disturbances after one year than those with lower levels of media use. This supports the notion that problematic use can be considered a trigger for psychopathological outcomes [49,50,51,52,53]. On the other hand, research shows that depressed people [20], for example, use their smartphones more to mitigate negative emotions. This supports the assumption that, conversely, psychopathological factors may trigger problematic use [54]. However, excessive use may in turn inadvertently lead to more stress and increased symptoms of psychopathology, resulting in a vicious cycle in which problematic use behavior and psychopathology interact [32].

Thus, Kardefelt-Winther [18] also argues in the context of the compensatory internet use theory that certain preferred internet applications can serve as compensators for increased feelings of stress, in that they initially lead to the feeling being regulated. Internet and smartphone use can therefore also be theorized as a coping tool whose effectiveness depends on certain conditions [55]. In the context of problematic smartphone use, it tends to be low; Zhao and Lapierre [56] examined relationships between coping strategies and feelings of stress. They found that people who prefer maladaptive strategies are more likely to view and use the internet as a stress-regulating medium to regulate their mood when faced with a stressful situation. Brand et al. [48] also found associations between maladaptive coping strategies and Internet addiction. Without interruption of these maladaptive mechanisms, a vicious cycle between psychopathology and problematic smartphone use can occur, such that increased feelings of stress lead to increased smartphone use, which then inadvertently further increases feelings of stress [20].

The so-called technostress is another aspect and refers to the stress that arises from the overload of information and communication due to modern technologies [57]. Lee et al. [37] found positive, significant associations between problematic smartphone use and technostress in a study. Therefore, it can be seen that the type of smartphone use can also have an impact on feelings of stress. For another example, Oberst et al. [58] found that excessive social media use in particular was associated with increased stress perception, and Murdock [59] describes the relationship between stress and the number of text messages that are written.

Thus, increased stress is not only a consequence of excessive and problematic smartphone use; the device or the type of use and of certain applications can also serve as a coping strategy to mitigate negative emotions and stress and be successful in doing so. Nevertheless, active coping and confrontation with the stress-triggering conditions remains absent, so that even more stress may result as a consequence.

### 1.5. Hypotheses

**Hypothesis** **1** **(H1).**
*Previous research confirms a link between problematic smartphone use and increased feelings of stress [19]. Some studies find problematic smartphone use as a cause and predictor of stress [31] as well as those that find increased stress as a cause and problematic smartphone use as a consequence [20], which then leads to more stress. It shall be demonstrated that a reciprocal relationship between problematic smartphone use and stress perception can be found.*


**Hypothesis** **2** **(H2).**
*A distinction is made between process, social, and habitual use [22,39]. Study results found either a higher influence of habitual and process use [22] or social use [23]. In this study it is to be shown that that the type of smartphone use has a positive impact on problematic use.*


**Hypothesis** **3** **(H3).**
*Research shows that smartphone use correlates with stress perception and can have both stress-inducing and stress-relieving effects [37]. It will be investigated whether the type of smartphone use (process, social, and habitual) also influences stress perception, as the different types of use serve different motives or gratifications of smartphone use [22]. It shall be demonstrated that the type of use has a positive impact on stress perception.*


**Hypothesis** **4** **(H4).**
*Finally, aspects of self-awareness of one’s smartphone use behavior will be examined. Taking into account the compensatory internet use theory [18], a person may be aware of existing problematic smartphone use and continue to perform it despite negative consequences. It is to be shown, the following hypotheses will be formulated: Perceptions of stress and problematic smartphone use will impact self-perceived problematic smartphone use behavior.*


## 2. Materials and Method

### 2.1. Participants and Procedure

A total of 147 people took part in the online survey (*N* = 147). After the exclusion of questionnaires that were not completely processed, 108 completely processed data sets remained for the evaluation (*N* = 108). Among the participants, 60.2% were women (*n* = 65), 38% were men (*n* = 41), and 1.9% were diverse (*n* = 2). The mean age was 31.8 years (*M_age_* = 31.8) with a standard deviation of 12.2 years (*SD_age_* = 12.2) with a minimum of 17 years (*Min* = 17) and a maximum of 70 years (*Max* = 70).

Professionally, the largest groups were employees (43.5%, *n* = 47) and students (35%, *n* = 38). A total of 7.4% (*n* = 8) were students, 5.6% (*n* = 6) indicated “other”, 4.6% (*n* = 5) were self-employed, 2.8% (*n* = 3) were retired, and one person was an employer*.

With regard to the level of education, 38.9% (*n* = 42) stated that they had a university degree, 38% (*n* = 41) had a high school diploma (German Abitur), 15.7% (*n* = 17) had completed vocational training, five people attended secondary school, and one person each had one of the German degrees of Volks-, Gesamt-, or Hauptschule, a Fachwirt, or no degree to date.

The present study uses the design of a cross-sectional observational study. The research project was advertised via a mailing list at the University of Koblenz-Landau and participation was encouraged. Convenience sampling was used for the study. This is the most commonly used sampling method [60] and was used for many other studies in the same research area [29,32,33]. Since the survey period took place during the COVID-19 pandemic and the accompanying social distance restrictions, this method was chosen to gather the largest possible number of subjects under the given conditions. The data collection was carried out online via the platform SoSciSurvey. The survey period was from 15 June–20 July 2021. The participation took approximately 20 min and the survey consisted of a questionnaire battery consisting of a sociodemographic questionnaire, the MPPUS [61], followed by the adapted scales used by van Deursen et al. [22] to determine the type of smartphone use, the Perceived Stress Scale (PSS) [62], and the self-assessed problem of smartphone use. All questions were asked in German.

### 2.2. Material

The materials used are standardized questionnaires that provide information about self-assessment. First, sociodemographic data (gender, age, occupation, education, and self-assessment of smartphone use and time on screen) were collected.

#### 2.2.1. MPPUS-10

To assess problematic smartphone use behavior (PSU), the short version of the Mobile Phone Problem Use Scale (MPPUS-10) by Foerster et al. (2015) [61] was used. Internal consistency is considered high with a Cronbach’s alpha > 0.8 [63]. The MPPUS was originally developed by Bianchi and Phillips (2005) [64] and contains 27 items. The MPPUS-10 consists of 10 questions answered on a 10-point Likert scale ranging from 1 = strongly disagree to 10 = strongly agree and describes various areas that are affected by smartphone use, such as escape from negative emotions or problems, cravings, development of tolerance, duration of use, negative effects on everyday life and the social and professional environment. A sum score is calculated from all items, this ranges from min = 10 to max = 100. There is no predetermined cutoff point at which smartphone use behavior is considered problematic; the scale is considered a continuum on which higher scores indicate a higher likelihood of problematic use [61,64].

#### 2.2.2. Types of Use

The type of use was asked using the items from the study by van Deursen et al. [22]. The English questions were translated into German for this study using the back-translation method [65]. The translated items can be seen in Appendix A. The type of use can be divided into three types of use: social use, process use, and habitual use. The types of use are asked in separate scales. Habitual smartphone behavior was measured using the habitual Internet use instrument first developed by Limayem, Hirt, and Cheung [40], which was adapted by van Deursen et al. [22] to measure habitual smartphone use. To determine the process and social use of the smartphone, van Deursen et al. [22] adapted a questionnaire by Chua, Goh, and Lee [66], which is used here.

The scales are each answered on a 5-point Likert scale from 1 = do not agree at all, to 5 = completely agree. Social use is surveyed with five items and a sum score is formed, ranging from min = 5 to max = 25. Internal consistency is considered questionable with a Cronbach’s alpha > 0.6 [63]. Process use is surveyed with seven items and a sum score is formed, ranging from min = 7 to max = 35. Internal consistency is found to be acceptable with a Cronbach’s alpha > 0.7 [63]. Habitual use is surveyed with six items and a sum score is formed, ranging from min = 6 to max = 30. The internal consistency can be classified as excellent with a Cronbach’s alpha > 0.9 [63]. Higher scores on the scales indicate a higher preference for the respective type of use [22].

#### 2.2.3. Perceived Stress Scale

To assess the perception of stress, the Perceived Stress Scale (PSS) [62], in the 10-item short version (PSS-10), was used. It measures in two subscales the degree to which the individual perceives situations in his or her life as uncontrollable and unpredictable. The subscale Perceived Helplessness has six items on a Likert scale from 1 = never, to 5 = very often. Internal consistency is considered excellent with a Cronbach’s alpha > 0.9 [63,67]. The subscale Perceived Self-Efficacy, has four items on a Likert scale from 1 = never, to 5 = very often. Internal consistency is considered high with a Cronbach’s alpha > 0.8 [63,67]. The sum score of the PH subscale and the reverse-coded items of the PSE subscale results in the total score and ranges from min = 10 to max = 50. Higher values represent a higher stress level.

#### 2.2.4. The Self-Assessed Problem of Smartphone Use

For the perception of one’s usage behavior, a question was created for self-assessment of the problematic nature of one’s smartphone usage behavior: “Do you generally perceive your smartphone usage behavior as problematic?” It is answered on a 5-point Likert scale from 1 = strongly disagree to 5 = strongly agree. The estimation of the reliability of scales is usually done by means of the calculation of coefficients of internal consistency. However, this approach can only be used for multi-item scales and not for single-item scales such as the one used in this study. For reliability estimation with single-item scales, procedures based on longitudinal data can be used. In this case, reliability is usually determined using the retest method, which is not possible here due to the cross-sectional design [68].

### 2.3. Statistical Analysis

The data analysis consists of descriptive statistics (means and standard deviations) and multivariate statistical analyses, primarily simple and multiple linear regressions on the variables PSU, PSS, types of use, and the self-assessed problem of smartphone usage behavior.

## 3. Results

Table 1 shows the mean values and standard deviations of the raw scores for the MPPUS, the PSS, and the subscales for the three subgroups of type of use.

In Table 2, the bivariate correlations between all variables used can be seen.

To answer H1, a Pearson correlation was calculated between problematic smartphone use and stress perception. The prerequisites for the analysis were audited and fulfilled. It was found that problematic smartphone use and stress perception correlate significantly with each other. The correlation shows a positive relationship between problematic smartphone use and stress perception (*r* = 0.416, *p* < 0.001, *N* = 108). According to Cohen [69], this corresponds to a medium effect. With a higher tendency to problematic smartphone use, stress perception also increases and the more the smartphone use behavior is considered problematic, the higher the stress perception is. H1 is confirmed.

Afterward, multiple linear regression analyses using the inclusion method and simple linear regressions were conducted. The prerequisites for all analyses were audited and fulfilled. Age and gender were included in the analyses as control variables.

Regression for H2 revealed that habitual use, process use and age were significant predictors of problematic smartphone use (Table 3). The predictors explained 49% of the variance in problematic smartphone use (*F*(5, 107) = 19.269, *p* < 0.001, *R*^2^ = 0.486), which corresponds to a high variance explanation [69]. H2 is confirmed.

For H3, linear regression analysis revealed that process use was a significant predictor of stress perception (Table 4). The predictor explained 18% of the variance in stress perception (*F*(5, 107) = 4.560 *p* < 0.001, *R*^2^ = 0.183), which corresponds to a low variance explanation [69]. H3 is confirmed.

To answer H4, a multiple linear regression analysis was calculated with problematic smartphone use and stress perception as predictors and the self-assessed problem of smartphone usage behavior as a criterion (Table 5). It was found that problematic smartphone use and stress perception have significant effects on the self-assessed problem of smartphone usage behavior (*F*(4, 107) = 17.441, *p* < 0.001, *R*^2^ = 0.404). The predictors explained 40% of the variance in self-assessed stress perception, corresponding to a middle effect [69]. H4 is confirmed.

## 4. Discussion

The present study dealt with the relationship between problematic smartphone use and stress perception and especially the question of whether the type of smartphone use behavior, namely process, social and habitual use, have an impact on problematic smartphone use and the related stress perception as well as shedding light on self-perceptions of one’s usage behavior. Here, it was shown that a relationship between problematic smartphone use and perceived stress can be found. The correlation found here confirms the results of previous studies already described [19,31] and shows that problematic smartphone use can have negative health consequences, as is made clear in this study using the example of stress perception. Explanations for this can be that people do not get to rest due to constant use, which can lead to an increased sense of stress. However, the results do not allow any conclusion as to which of the variables influences the relationship, which is why a reciprocal relationship is assumed [20]. Accordingly, an increased perception of stress can equally lead to problematic smartphone use. This is in line with the compensatory internet use theory, according to which the individual sees internet or smartphone use as a solution to problems, i.e., ending negative emotions and contributing to distraction [18].

Additionally, habitual use and process use and age were shown to be predictors of problematic smartphone use. However, habitual use has the largest effect, followed by process use while age has the smallest effect on problematic smartphone use. This result is in line with existing research findings, as habitual use is strongly related to dependent behavior and for this reason, an automatism is formed towards habitual use [22]. Process use has only a slightly smaller effect. One possible reason for this result could be that this form of use focuses on personal pleasure and the associated satisfaction without a higher benefit (watching funny videos or memes, playing a game) [19], as described in the uses-and-gratifications approach, which is why it tends to be perceived as procrastination and thus disruptive in modern achievement societies. That age has an effect on PSU has already been shown in previous studies, e.g., with the help of generational comparisons [32]. Social use and gender have no significant effect on PSU. For social use, it may be related to the fact that this type of use is about contact and interaction with other people and is therefore considered less disruptive and more helpful regarding the compensatory-internet-use-approach [18]. This becomes particularly relevant when considering that the study presented here was collected during the COVID-19 pandemic and the associated contact restrictions and that social contact was therefore often shifted to the smartphone and this was the only form of social exchange.

For stress perception, only process use is a predictor. This result can also be well explained by the fact that it is merely a matter of one’s pleasure and short-lived distraction, and that this represents a form of procrastination in the sense of striving for performance, which leads to a guilty conscience and thus a higher sense of stress. Concerning social use, one would have expected an effect, since it is assumed that social exchange can also be considered a stress reliever [18,20]. Reasons for the non-significant result could be the low internal consistency of the scale, which is questionable for this sample, but is reported higher in the original study, which is why the scale was used here. Another reason might be the fact that the current COVID-19 pandemic has increased the general stress level to such an extent that coping mechanisms that are commonly used, such as social exchange in person, is now limited to the smartphone, and therefore no longer as effective as before, because digital exchanges cannot adequately replace real ones [18,70]. Since habitual use is an automatism and therefore an unconscious behavior, it can be assumed that this leads to less stress perception in the sense of the action regulation theory [71] since these are subordinate processes.

Both problematic smartphone use and stress perception are predictors of self-perceived problematic smartphone use. Based on these results, it can be assumed that most people are aware of the increasingly problematic nature of their usage behavior. However, as could be seen from the other hypotheses, it is evident that, despite this awareness of one’s own problematic use, no change in behavior occurs. Process use in particular has been shown in this study to be a factor in stress perception, which is why it is important not only to be aware of one’s problematic behavior, but also to actively change it, for example by reading fewer online amounts, playing in-app games, or consuming videos, in order to develop a healthier and, above all, less stressful approach to smartphone use.

### Limitations

The study presented here has a few limitations. First of all, the survey was based upon self-reports, which could have led to a biased set of answers. In addition, the subscale of social use in this sample showed only questionable internal consistency. It must also be considered that the sample size for a cross-sectional study could be larger, as measurement errors may have a greater impact in smaller samples [72]. Beyond this, it should be noted that the constructs examined here are each interdependent. The reciprocal relationship between the constructs of problematic smartphone use and factors such as stress has already been demonstrated in previous studies [20,54], but must also be mentioned again here, since the possibility of reverse causation also exists in this study.

Furthermore, it must be taken into account that the study was conducted at the time of the COVID-19 pandemic, which led to a strong overall increase in both smartphone use and general stress perception. An influence of possible associated consequences and changes in media use behavior of individuals was not considered in this study and may lead to different results in future studies. Future studies should focus on prospective longitudinal surveys to show representative results on cause and effect relationships. Furthermore, to the possibility of longitudinal study, experiments should also be used as a research design for future studies in which, for example, manipulation could be established by abstinence from smartphone use. This would provide important inferences about the causality and direction of effect of the constructs [73,74]. In addition, other variables besides stress perception should be included in the study. For example, the quality of sleep or the quality of relationships with others could be investigated as significant factors as well as psychophysiological and biochemical indicators of stress and addiction.

## 5. Conclusions

In particular, this study was able to show that a higher risk of PSU as well as certain types of use, especially the process use of the smartphone, can be associated with an increased sense of stress. At the same time, a large proportion of people are aware of the potential problems associated with their usage behavior, but at the same time are not willing to change anything about it—as a result, they are unaware of the risks and the scientifically proven consequences of excessive smartphone use on the psyche (increase in depression, anxiety disorders, eating disorders, etc. [19,29]) and, as has been shown here, on their own perception of stress.

For this very reason, it is important to continue to educate people about the warning signs and health consequences of possible problematic smartphone use. This also includes imparting knowledge about stress management strategies and, in particular, about the fact that distraction by the smartphone may have a stress-relieving effect in the short term, but is not very effective in the medium and long term and may even lead to an increased perception of stress [20]. With this information in mind and an understanding of the risk factors, the smartphone is used to be a wholesome enrichment and simplification of everyday life without overuse or problematic addiction and increased perception of stress.

## Figures and Tables

**Table 1 behavsci-12-00099-t001:** Descriptive statistics for MPPUS, PSS, and type of use (social use, process use, and habitual use).

	Mean	SD
PSU	43.15	15.68
PSS	30.02	7.70
Social use	20.42	3.14
Process use	25.24	4.33
Habitual use	25.65	4.17

**Table 2 behavsci-12-00099-t002:** Bivariate correlations after Pearson with MPPUS, PSS, self-assessed problem use behavior, and type of use (Social use, process use, and habitual use).

	1	2	3	4	5	6
PSS						
PSU	0.416 **					
Self-assessed problem use	0.400 **	0.611 **				
Process use	0.371 **	0.560 **	0.418 **			
Social use	0.080	0.275 **	0.152	0.164		
Habitual use	0.297 **	0.580 **	0.489 **	0.546 **	0.167	
Age	−0.262 **	−0.405 **	−0.271 **	−0.311	−0.210	−0.216

** indicates *p* < 0.01.

**Table 3 behavsci-12-00099-t003:** Predictors of PSU (*N* = 108).

Type of Use	ß	*t*	*p*	*R* ^2^
Process use	0.275	−3.144	0.002	0.486
Social use	0.125	1.701	0.092	
Habitual use	0.364	4.264	<0.001	
Age	−0.216	−2.836	0.006	
Gender	−0.007	−0.102	0.919	

**Table 4 behavsci-12-00099-t004:** Predictors of stress perception (*N* = 108).

Type of Use	ß	*t*	*p*	*R* ^2^
Process use	0.253	2.298	0.024	0.183
Social use	−0.021	−0.227	0.821	
Habitual use	0.122	1.131	0.261	
Age	−0.153	−1.599	0.113	
Gender	0.099	1.099	0.275	

**Table 5 behavsci-12-00099-t005:** Predictors of “Self-assessed problem of smartphone usage behavior” (*N* = 108).

Predictors	ß	*t*	*p*	*R* ^2^
PSU	0.534	6.013	<0.001	0.399
Stress perception	0.184	2.172	0.032	
Age	−0.001	−0.151	0.880	
Gender	−0.152	−0.867	0.388	

## Data Availability

The data used to analyze the results reported here are available for public review at https://mfr.osf.io/render?url=https%3A%2F%2Fosf.io%2Fnb8ks%2Fdownload.

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
