# Peer review of "Does the Type of Smartphone Usage Behavior Influence Problematic Smartphone Use and the Related Stress Perception?"

_behavsci, 2022, doi:10.3390/bs12040099_

Round 1
Reviewer 1 Report
The topic addressed by the manuscript is an interesting and very topical topic, in addition to being innovative, so the effort to make the introduction and justification is important and the authors have addressed it correctly.
However, with respect to the methodological approach, there are different fissures.
The first is the content used to make the hypotheses, I believe that the hypotheses are null or alternative should be concise and be aimed at proving what the research wants, not to justify the content of it.
On the other hand, the authors do not specify the design used to carry out the methodological approach, it seems to be a cross-sectional observational study, but they do not define it.
When the authors refer: "The study was advertised via a mailing list at the University of Koblenz-Landau and 224 participation was encouraged. The data collection was carried out online via the platform 225 SoSciSurvey. The survey period was from 06/15 - 07/20/2021. The participation took ap-226 proximately 20 minutes and the survey consisted of a questionnaire battery consisting of 227 a sociodemographic questionnaire, the MPPUS [60], followed by the adapted scales used 228 by van Deursen et al. [22] to determine the type of smartphone use, the Perceived Stress 229 Scale (PSS) [61], and the self-assessed problem of smartphone use. All questions were 230 asked in German". They are referring that they make an indiscriminate selection of the sample, this type of sampling that has materialized since the pandemic, have a scientific validity if the procedure and intentionality of it is described, as well as references of similar studies that carried it out and none of that appears.
As for the data collection instruments used, all are validated except for their own that is not specified during the procedure how they were going to integrate it into the analysis of the same, since it is something not validated and carried out by the group of authors.
"The self-assessed problem of smartphone use 283
For the perception of one's usage behavior, a question was created for self-assessment 284 of the problematic nature of one's smartphone usage behavior: "Do you generally perceive 285 your smartphone usage behavior as problematic?" It is answered on a 5-point Likert scale 286 from 1 = strongly disagree to 5 = strongly agree. 287"
The authors describe the methodology of data selection and collection as limitations, so it is difficult to improve these methodological fissures that I have described above.
Greetings
Author Response
Review 1
Thank you very much for the critical review of the study and the good comments for improvement! I really appreciate your input and thanks for taking the time to conduct your feedback. According to your expertise, I have changed or adapted the following points.
The topic addressed by the manuscript is an interesting and very topical topic, in addition to being innovative, so the effort to make the introduction and justification is important and the authors have addressed it correctly.
Thank you very much for that feedback, it is really appreciated.
However, with respect to the methodological approach, there are different fissures. The first is the content used to make the hypotheses, I believe that the hypotheses are null or alternative should be concise and be aimed at proving what the research wants, not to justify the content of it.
Thank you for pointing out that the hypotheses were not sufficiently elaborated. This has been changed as follows:
Previous research confirms a link between problematic smartphone use and increased feelings of stress [19]. Some studies find problematic smartphone use as a cause and predictor of stress [31] as well as those that find increased stress as a cause and problematic smartphone use as a consequence [20] which then leads to more stress. It shall be demonstrated that a reciprocal relationship between problematic smartphone use and stress perception can be found (H1).
A distinction is made between process, social, and habitual use [22,39]. Study results found either a higher influence of habitual and process use [22] or social use [23]. In this study it is to be shown that that the type of smartphone use can have a positive impact on problematic use (H2).
Research shows that smartphone use correlates with stress perception and can have both stress-inducing and stress-relieving effects [37]. It will be investigated whether the type of smartphone use (process, social, and habitual) also influences stress perception, as the different types of use serve different motives or gratifications of smartphone use [22]. It shall be demonstrated that the type of use has a positive impact on stress perception (H3)
Finally, aspects of self-awareness of one's smartphone use behavior will be examined. Taking into account the compensatory internet use theory [18], a person may be aware of existing problematic smartphone use and continue to perform it despite negative con-sequences. It is to be shown, the following hypotheses will be formulated: Perceptions of stress and problematic smartphone use will impact self-perceived problematic smartphone use behavior (H4).
On the other hand, the authors do not specify the design used to carry out the methodological approach, it seems to be a cross-sectional observational study, but they do not define it.
Thank you for the comment that it was not made clear enough what kind of design was used. According to your expertise, we added a sentence to the research design.
The present study uses the design of a cross-sectional observational study.
When the authors refer: "The study was advertised via a mailing list at the University of Koblenz-Landau and participation was encouraged. The data collection was carried out online via the platform SoSciSurvey.
The survey period was from 06/15 - 07/20/2021. The participation took approximately 20 minutes and the survey consisted of a questionnaire battery consisting of a sociodemographic questionnaire, the MPPUS [60], followed by the adapted scales used by van Deursen et al. [22] to determine the type of smartphone use, the Perceived Stress Scale (PSS) [61], and the self-assessed problem of smartphone use. All questions were asked in German". They are referring that they make an indiscriminate selection of the sample, this type of sampling that has materialized since the pandemic, have a scientific validity if the procedure and intentionality of it is described, as well as references of similar studies that carried it out and none of that appears.
Thank you for the valuable advice, which we have gladly implemented and explained the sample recruitment method in more detail:
…The study was advertised via a mailing list at the University of Koblenz-Landau and participation was encouraged. Convenience sampling was used for the study. This is the most commonly used sampling method (Acharya, A. S., Prakash, A., Saxena, P., & Nigam, A. (2013). Sampling: why and how of it. Indian Journal of Medical Specialties, 4(2), 330-333.) and was used for many other studies in the same research area [29, 32, 33]. Since the survey period took place during the Covid-19 pandemic and the accompanying social distance restrictions, this method was chosen to gather the largest possible number of subjects under the given conditions.
As for the data collection instruments used, all are validated except for their own that is not specified during the procedure how they were going to integrate it into the analysis of the same, since it is something not validated and carried out by the group of authors.
"The self-assessed problem of smartphone use
For the perception of one's usage behavior, a question was created for self-assessment of the problematic nature of one's smartphone usage behavior: "Do you generally perceive your smartphone usage behavior as problematic?" It is answered on a 5-point Likert scale from 1 = strongly disagree to 5 = strongly agree."
Thank you for this useful critique! You are absolutely right that no information is provided to validate the scale. Since this is the first use of the scale and one-item scales can only be validated longitudinally, we must refrain from providing further information on this here. However, thanks to your helpful advice, we have actively made this clear in the text:
The estimation of the reliability of scales is usually done by means of the calculation of coefficients of internal consistency. However, this approach can only be used for multi-item scales and not for single-item scales such as the one used in this study. For reliability estimation with single-item scales, procedures based on longitudinal data can be used. In this case, reliability is usually determined using the retest method, which is not possible here due to the cross-sectional design (Schermelleh-Engel & Werner, 2012).
The authors describe the methodology of data selection and collection as limitations, so it is difficult to improve these methodological fissures that I have described above.
Thank you again for the good advice, which we gladly accepted and implemented according to your expertise to enhance the study and eliminate possible misunderstandings.
We hope that with the newly introduced improvements, the methodology of data selection and collection is no longer a limitation. Since, for example, the use of a convenience sample is one of the most commonly chosen methods (Acharya et al., 2013) and, during the Covid-19 restrictions, it provided one of the few opportunities to continue the scientific work in order to be able to identify new scientific trends.
Literature:
Acharya, A. S., Prakash, A., Saxena, P., & Nigam, A. (2013). Sampling: why and how of it. Indian Journal of Medical Specialties, 4(2), 330-333.
Greetings
Greetings
Reviewer 2 Report
This is a very interesting study examining whether different type of smartphone usage behavior predict smartphone use and perceived stress. The paper is well-written and all the analyses were carefully conducted. I personally believe that this study is promising and will have important contribution to the literature. While most existing studies are focusing on total smartphone or other technological devices screen time (e.g., video gaming), it is commendable that the authors delved deeper and focused on context surrounding technology use. This is in line with the most recent direction and suggestion in the literature
Busch, P. A., & McCarthy, S. (2021). Antecedents and consequences of problematic smartphone use: A systematic literature review of an emerging research area. Computers in Human Behavior, 114, 106414.
I only have a few comments to improve the manuscript further:
- The authors should describe the sociodemographic information of their participants in the method section beyond age and gender (e.g., occupation, education). These can be integrated within Table 1.
- The authors should consider to control for demographics in their regression analyses to examine whether the results are robust with adjustment.
- The sample size is lower than expected for a cross-sectional study. The authors should acknowledge this in their limitation.
Loken, E., & Gelman, A. (2017). Measurement error and the replication crisis. Science, 355(6325), 584-585.
- Another limitation of the current study is the possibility of reverse causation. It is highly possible that stress can also affect smartphone usage, and vice versa. It will be important for the authors to discuss this possibility in their limitation and provide more discussion. For future direction, other than longitudinal studies, it will be useful for the authors to highlight the possibility to conduct experiment to causal relationship. One way to manipulate smartphone usage would be by manipulating smartphone abstinence (see relevant references below):
Hartanto, A., Quek, F. Y., Tng, G. Y., & Yong, J. C. (2021). Does social media use increase depressive symptoms? A reverse causation perspective. Frontiers in Psychiatry, 12, 335.
Valkenburg, P. M. (2021). Social media use and well-being: What we know and what we need to know. Current Opinion in Psychology.
Author Response
Review 2
Thank you very much for the critical review of the study and the good comments for improvement! I really appreciate your input and thanks for taking the time to conduct your feedback. According to your expertise, I have changed or adapted the following points.
This is a very interesting study examining whether different type of smartphone usage behavior predict smartphone use and perceived stress. The paper is well-written and all the analyses were carefully conducted. I personally believe that this study is promising and will have important contribution to the literature. While most existing studies are focusing on total smartphone or other technological devices screen time (e.g., video gaming), it is commendable that the authors delved deeper and focused on context surrounding technology use. This is in line with the most recent direction and suggestion in the literature
Busch, P. A., & McCarthy, S. (2021). Antecedents and consequences of problematic smartphone use: A systematic literature review of an emerging research area. Computers in Human Behavior, 114, 106414.
Thank you very much for that feedback, it is really appreciated.
I only have a few comments to improve the manuscript further:
The authors should describe the sociodemographic information of their participants in the method section beyond age and gender (e.g., occupation, education). These can be integrated within Table 1.
Thank you for pointing out that the sociodemographic information was not sufficiently elaborated. This has now been integrated into the text as follows:
Professionally, the largest groups were employees (43.5%, n = 47) and students (35%, n = 38). 7.4% (n = 8) were students, 5.6% (n = 6) indicated "other", there were 4.6% (n = 5) self-employed, 2.8% (n = 3) retired, one person is an employer*.
With regard to the level of education, 38.9% (n = 42) stated that they had a university degree, 38% (n = 41) had a high school diploma (German Abitur), 15.7% (n = 17) had completed vocational training, five people attended secondary school, and one person each had one of the German degrees of Volks-, Gesamt- or Hauptschule, a Fachwirt, or no degree to date.
Since only the mean values and standard deviations of the used scales were described in Table 1, the explanations of the sociodemographic data were described separately to avoid affecting clarity of the table, especially since the German school system has many differentiations, which makes it more complicated.
The authors should consider to control for demographics in their regression analyses to examine whether the results are robust with adjustment.
Thanks for this advice. We added age and gender in the regression analyses:
Afterward, multiple linear regression analyses using the inclusion method and simple linear regressions were conducted. The prerequisites for all analyses were audited and fulfilled. Age and gender were included in the analyses as control variables.
Regression for H2 revealed that habitual use, process use and age were significant predictors of problematic smartphone use (Table 3). The predictors explained 49% of the variance in problematic smartphone use (F(5, 107) = 19.269, p < .001, R2 = .486), which corresponds to a high variance explanation [67]. H2 is confirmed.
Table 3. Predictors of PSU (N = 108)
|
ß |
t |
p |
R2 |
|
|
Type of use |
||||
|
Process use |
.275 |
-3.144 |
.002 |
.486 |
|
Social use |
.125 |
1.701 |
.092 |
|
|
Habitual use |
.364 |
4.264 |
< .001 |
|
|
Age |
-.216 |
-2.836 |
.006 |
|
|
Gender |
-.007 |
-.102 |
.919 |
|
For H3, linear regression analysis revealed that Process use was a significant predictor of stress perception (Table 4). The predictor explained 18% of the variance in stress perception (F(5, 107) = 4,560 p < .001, R2 = .183), which corresponds to a low variance explanation [67]. H3 is confirmed.
Table 4. Predictors of stress perception (N = 108)
|
|
ß |
t |
p |
R2 |
|
Type of use |
|
|
|
|
|
Process use |
.253 |
2.298 |
.024 |
.183 |
|
Social use |
-.021 |
-.227 |
.821 |
|
|
Habitual use |
.122 |
1.131 |
.261 |
|
|
Age Gender |
-.153 .099 |
-1.599 1.099 |
.113 .275 |
|
To answer H4, a multiple linear regression analysis was calculated with problematic smartphone use and stress perception as predictors and the self-assessed problem of smartphone usage behavior as a criterion (Table 5). It was found that problematic smartphone use and stress perception have significant effects on the self-assessed problem of smartphone usage behavior (F(4, 107) = 17.441, p < .001, R2 = .404). The predictors explained 40 % of the variance in self-assessed stress perception, corresponding to a middle effect [67]. H4 is confirmed.
Table 5. Predictors of “Self-assessed problem of smartphone usage behavior” (N = 108)
|
|
ß |
t |
p |
R2 |
|
Predictors |
|
|
|
|
|
PSU |
.534 |
6.013 |
< .001 |
.399 |
|
Stress perception |
.184 |
2.172 |
.032 |
|
|
Age |
-.001 |
-.151 |
.880 |
|
|
Gender |
-.152 |
-.867 |
.388 |
|
The sample size is lower than expected for a cross-sectional study. The authors should acknowledge this in their limitation. (Loken, E., & Gelman, A. (2017). Measurement error and the replication crisis. Science, 355(6325), 584-585.)
Thanks for the good advice and the interesting article on this. According to your expertise, the sample size was included in the limitation as follows and the source was cited:
It must also be considered that the sample size for a cross-sectional study could be larger, as measurement errors may have a greater impact in smaller samples [70].
Another limitation of the current study is the possibility of reverse causation. It is highly possible that stress can also affect smartphone usage, and vice versa. It will be important for the authors to discuss this possibility in their limitation and provide more discussion.
Thank you for this good advice, which we have gladly implemented and clarified the reciprocal effect of the constructs in the limitations as follows:
Beyond this, it should be noted that the constructs examined here are each interdependent. The reciprocal relationship between the constructs of problematic smartphone use and factors such as stress has already been demonstrated in previous studies [20, 54], but must also be mentioned again here, since the possibility of reverse causation also exists in this study.
For future direction, other than longitudinal studies, it will be useful for the authors to highlight the possibility to conduct experiment to causal relationship. One way to manipulate smartphone usage would be by manipulating smartphone abstinence (Hartanto, A., Quek, F. Y., Tng, G. Y., & Yong, J. C. (2021). Does social media use increase depressive symptoms? A reverse causation perspective. Frontiers in Psychiatry, 12, 335.; Valkenburg, P. M. (2021). Social media use and well-being: What we know and what we need to know. Current Opinion in Psychology.)
Thank you for this good idea, which we have gladly included in the outlook for future studies:
Furthermore, to the possibility of longitudinal study, experiments should be used as a research design for future studies in which, for example, manipulation could be established by abstinence from smartphone use. This would provide important inferences about the causality and direction of effect of the constructs [71, 72].
Round 2
Reviewer 2 Report
The authors have addressed all my concerns and comments well.